# Behavioral and psychosocial factors of quality of life among adult people living with HIV on Highly Active Antiretroviral Therapy, in public hospitals of Southwest Ethiopia

**Abreha Addis Gesese** [1] *, **Yitages Getachew Desta**[2], **Endale Zenebe Behire** [3]

1 Department of Clinical Nursing, Gambella Teachers Education and Health Science College, Gambella Town, Gambella, South West Ethiopia, 2 Department of Environmental Health Science and Technology, Gambella Teachers Education and Health Science College, Gambella Town, Gambella, South West Ethiopia, 3 Department of Epidemiology, Faculty of Public Health, Jimma University, Jimma, Ethiopia

* abrhaddis09@gmail.com

## Abstract

Despite the availability of Highly Active Antiretroviral Therapy, the quality of life (QOL) of People Living with HIV/AIDS (PLWHIV) has continued to be affected. However, previous studies focused on the magnitude and clinical determinants which lacks behavioral and psychosocial factors of QOL. Thus, this study aimed to identify behavioral and psychosocial determinants of QOL among Adult PLWHIV on HARRT, in Public Hospitals of Jimma Zone, South West, Ethiopia, 2018. A cross-sectional study design was conducted in Public Hospitals of Jimma Zone, Southwest Ethiopia from March 10 to April 10/2018. QOL of was measured using WHOQOL-HIV BREF instrument. A simple random sampling technique was employed to enroll study participants. A pretested interviewer-administered structured questionnaire was used to collect data. Then, data were entered into Epi-Data version 3.1 and analyzed using SPSS version 20. Bivariate and multiple variable logistic regression analyses were also performed. A total of 300 respondents were enrolled into the study yielding a response rate of 97.7%. The majority of respondents were from urban residence and between 35–44 years of age. About 47% of respondents have ever used substances, and 58.3% have obtained social support. Nearly 80% and 26.3% of the study participants were stigmatized and severely depressed. More than half of the study participants had good overall QoL with the highest domain QOL in level of independence and lowest in social relations. Factors associated with poor physical health include being government employee AOR 0.33 95%CI (0.15, 0.69), from private business AOR 0.33 95%CI (0.14, 0.79), being 1st wealth quintile AOR 2.44 95%CI (1.16, 5.14), and not obtaining financial support AOR 4.27 95%CI (1.94, 9.42). Lower wealth index has been associated with almost all domain scores of poor QOL except spiritual domain. More than half of the respondents had good overall QoL with the highest domain score in level of independence and lowest in social relations domain. Several factors have contributed to poor domain QOL of PLWHIV. Therefore, it will become all the most important to develop effective strategies, policies and programs targeting people living with HIV. Emphasis should be given to the socio-economic factors that affect their

**Data Availability Statement:** All relevant data are within the paper and its Supporting Information files.

**Funding:** The authors received no specific funding for this work.

**Competing interests:** The authors declare that they do not have competing interest.

QOL on HAART. Professional counseling and guidance with life skill packages should be strengthened to cope up with adverse behavioral factors. Finally, psychosocial support should be provided from all responsible bodies.

## Background

HIV/AIDS has been a fatal illness that causes life-threatening opportunistic infections, neurological disorders, and unusual malignancies with a public health concern that has progressed from mysterious illness to global pandemic [1, 2]. The Sub Saharan Africa region constitutes more than half of the People living with HIV/AIDS. Ethiopia is among Sub-Saharan Africa with highly burdened areas of disease prevalence of 1.1% with a higher distribution in urban and peri-urban areas [3, 4]. HIV/AIDS has shifted from an acutely fatal disease to a chronic manageable condition due to the commitment of biopharmaceutical researchers, health care workers, and advocates [5]. Introduction of highly active antiretroviral therapy (HAART) has become the cornerstone of clinical intervention to prevent transmission and slow progression of HIV infection in individuals living with HIV/AIDS [6, 7]. It also helps to live longer and manages properly with diagnosis and treatment. It helps to reduce HIV-related morbidity and mortality, provide maximal and durable suppression of viral load (VL), and restore and/or preserve immune function [8–10]. Approximately 43%, of the annual number of people dying from AIDS-related causes has declined because of access to HAART reaching 54% among adults aged 15 years and above since 2003which also improved quality of life [2, 11].

Quality of life is, therefore, an important outcome measure used in a wide variety of medical research to ascertain aspects of well-being in settings of health and disease. It can be useful for understanding the life experience of PLWHIV which will become more important in evaluating their status and providing necessary health and social services [11]. It is defined differently by different organizations. Communicable Disease Control (CDC) defined Quality of life (QOL) as an overall sense of well-being, including aspects of happiness and satisfaction with life as a whole. It is broad and subjective rather than specific and objective [12]. However, commonly used measures in clinical practice include items that clinicians consider relevant to PLWHIV as the use of objective measures which include the severity of symptoms and level of functioning [13]. They also provide a basis for sharing clinical decision-making between patients and clinicians, identifying patients' priorities for treatment, and facilitating the setting of realistic treatment goals [14]. The WHO defines quality of life as individuals' perceptions of their position in life in the context of the culture and value systems in which they live and in relation to their goals, expectations, standards, and concerns. It refers to a subjective evaluation that focuses on respondents' "perceived" quality of life [15].

Despite the different strategies provoked and tremendous efforts have been made to halt the spread and consequences of HIV/AIDS both at the global and national levels, PLWHIV continued to develop a poor quality of life. PLWHIV can be affected in a complex way by the person's physical health, psychological state, level of independence, social relationships, personal beliefs, and their relationship to salient features of their environment [15]. The Quality of Life (QOL) increases with ART [16]. The World Health Organization report of 2015, persons infected and affected with HIV and livelihood hazards are often subjected to social stigma and discrimination in the workplace and in some cases forced to leave or terminate employment especially in resource-poor settings. Low- and Middle-Income Countries revealed that social support is associated with improvement in access and adherence to ART, medication uptake, and retention in care, physical functioning, CD4 cell progression, virologic

suppression, bodyweight of PLWHA, and mortality with improvements in psychosocial functioning [2, 7, 11, 17].

Studies show that HIV/AIDS continues to affect individuals physically, mentally, socially, and financially with a great impact on society both as an illness and as a source of discrimination and stigma [2, 15]. It has also been associated in a complex way with impoverished housing conditions and hindering access to PLHIV and affected communities to work and treat. The challenges of slums in urban areas such as basic services, inadequate water, sanitation, overcrowding, and others are worsened by the impact of HIV/AIDS [18, 19]. According to a report from UNAIDS in Ethiopia People who report that they would not buy vegetables from a shopkeeper living with HIV (59.9%) [20]. A study from Bahir-dar revealed that more than half had a low QOL and a study from Mekele showed significant gender differences with lower HRQOL in females in almost all domains [21, 22].

A number of studies have been conducted on the quality of life of people living with HIV/ AIDS [23–30]. Good nutrition is essential for achieving and preserving health while helping the body protect itself from infection. Therefore, it can improve the quality of life of PLHIV and help them live longer [31]. The study took place in an area where high numbers of PLWHIV were on ART. The Jimma zone is boarded by the Gambella Region, in Southwestern Ethiopia, which accounts for the highest HIV prevalence in Ethiopia as evidenced by a high number of travelers crossing the Gambella region known for its refugees and mining workers who put the town at high risk of contracting the disease [3, 32]. Given the benefits of measuring patients' treatment outcomes on HAART; there is a paucity of research on behavioral and psychosocial factors of quality of life in which, most of the studies were cross-sectional in nature and focused on clinical and nutritional determinants [33–36]. With the demanding and significant benefit of monitoring and evaluation of patients' outcomes on HAART, this study aimed to identify behavioral and psychosocial factors affecting the quality of life among PLWHIV on HAART in the selected study setting of Jimma Zone.

## Methods and materials

### Study design, method and population

A cross sectional study design was conducted in public hospitals of Jimma Zone, Southwest Ethiopia from March 10 to April 30/2018. The Jimma zone is located 357 km southwest of Addis Ababa. This zone is found in a region that accounts for the highest number of HIV-infected people in Ethiopia. It is found near the Gambella region, a region that accounts for the highest prevalence rate of HIV from Ethiopia [3]. The study was conducted in four public hospitals called Jimma University Medical Center (JUMC), Shenen Gibe Hospital (SGH), Agaro hospital (AH), and Limu Genet Hospital (LGH). These were selected due to the provision of ART services for a longer period of time, having many patients on ART and their electronic medical records. These health institutions serve a catchment area of approximately 3 million people in Southwest Ethiopia. A total of 11,500 PLWHA were followed up in HIV chronic care during the study period. All adult PLWH (aged greater than or equal to 15 years) enrolled in HAART at public hospitals of Jimma Zone were the source population. All adult PLWH (aged greater than or equal to 15 years) attending HAART during the study period at the study settings were included as study population. Severely ill participants who were unable to respond and those participants with incomplete charts were excluded.

### Sample size determination and sampling technique

The sample size was calculated by using a single population proportion formula using 76.2% proportion of good quality of life among PLWHIV by considering the following parameters:

95% Confidence Level, 5% of maximum discrepancy between the upper and lower sample size. By adding 10% non-response rate, the total calculated sample size was 307 participants [37]. A simple random sampling technique was used to recruit study participants after the sample was proportional allocation to the size of each hospital.

## Data collection instruments and procedures

A pretested structured questionnaire comprising of WHOQOL-HIV BREF instrument with six dimensions of QOL assessment tool which includes measuring physical dimension, psychological dimension, level of independence, social relations; environmental and religious/ spiritual/personal beliefs that help identify advanced disease states with recurrent infections, including socio-demographic, economic, behavioral, and psychosocial was prepared and used to measure QOL among study participants. The items were contextualized to the study area and translated into the local language (Amharic and Affan Oromo) using forward and backward translation. The reliability and validity of the tool were ensured and used by other researchers in Ethiopia, including the study areas [18, 21, 22, 25].

A face-to-face interview using an interviewer- administered tool was collected by two ART nurses in each hospital. One supervisor was assigned to each site to oversee the process. The principal investigator supervised every aspect of the data collection process along with supervisors and the ART nurses mentored shouldering the regular data collection task. The filled questionnaire was gathered by the principal investigator and/or the supervisor on a daily basis.

## Data processing and analysis

Collected data were first checked manually for completeness and consistency by supervisors during the time of data collection and rechecked again at the office by the principal investigator before data entry. Then, data were entered into Epi Data version 3.1 and exported to SPSS version 20 for analysis. Descriptive analysis was carried out and summarized by narration, tables and graphs. The Principal component analysis was performed for wealth index, stigma, and depression score [38]. Cronbach's alpha was used to test for internal consistency and reliability of PCA and accepted above 0.7 and the depression was measured and its score was accepted at a scale-reliability of 0.935. Domain scores in the WHOQOL-HIV were scaled in positive direction with higher score denoting good quality of life. Negative questions like pain and discomfort were recorded so that higher scores reflected better QOL. Mean scores of the items within each domain were used to calculate the domain score. Mean scores were then multiplied by 4 in order to make domain score comparable with the scores used in WHO-QOL-100 [15]. By taking the mean scores of each domain and the overall QoL as a cutoff point, QOL was dichotomized as poor or good. Individuals who scored below the mean were classified as having poor quality of life on each of the six domains.

The variables with p-values<0.05 in the final model declared statistically significant association. To assess predictors of QOL (poor versus good), first univariate analysis was employed and then variables that show statistically significant association with each of the six WHOQOL domains in the univariate analysis ($P$ <0.25) were entered into a multiple variable logistic regression model. A significance level was set at $P$ < 0.05. Finally, goodness of fit of the final model was checked using *Hosmer* and *Lemeshow* statistic.

## Data quality management

The principal investigator used a pre-tested questionnaire on 5% of respondents prior to the actual data collection in Bedele General Hospital and amendments were made. Local languages were used to get valid data. Data collectors and supervisors were given training on the data

collection tool contents, how to collect data and to make the question items understood uniformly. During data collection, the principal investigator ensured that the collected data fulfills the expected procedures and kept every question responded to properly through spot-checking. When data collectors face problems during interviews, supervisors, as well as the principal investigator, actively support them. After data editing, coding, and entry were made, data cleaning was taken place to check for the consistency of data and identify errors that occur during data collection or coding process.

## Ethical approval and consent to participate

Ethical clearance was obtained from Jimma University Research Ethics Review Committee. A formal written letter was granted from the Population and Family health department, and given to Jimma zone health office and the respective hospitals. Permission was obtained from each facilities and written informed consent was obtained from all informants including guardians whose child was under the age of 18 years. The interview procedure was conducted completely in a private room. The informants were ensured that all the data would be kept confidential by using codes to identify participants. The Participants were clearly informed about their right to refuse to participate in the study or withdraw at any time during the interview session.

## Result

### Socio-demographic and economic characteristics of respondents

A total of 300 respondents were enrolled into the study yielding a response rate of 97.7%. The majority of respondents were from urban residence and 187(62.3%) were female in sex category. Beside this, 133(44.3%) were between ages 35–44 years of age and 35.7% were less than or equal to grade four in educational status. The rest 39%, 55%, 60.3%, and 34.7% were from private business, Oromo in ethnicity, single in marital status, and in the second wealth quintile, respectively (**Table 1**).

### Behavioral related characteristics of respondents

This study found that 46.7% of respondents have ever used any substance. 18.7% and 32.3% of respondents have ever drunk alcohol and chewed khat respectively. However, in their current status 89% and 76.3% of respondents never drink alcohol and chewed khat respectively. 76.3% of respondents have disclosed their HIV/AIDS status (**Table 2**).

### Psychosocial characteristics of respondents

This study found that 58.3% of respondents obtained support. 24% of respondents obtained emotional support, 20% financial support, and 26% physical care and support. The majority of respondents obtained social support from their families however, 6.3% of the support was obtained from friends, 4.7% was from NGOs and 40.7% of respondents were satisfied with the support obtained from different sources. Nearly 80% and 26.3% of the study participants were stigmatized and severely depressed (**Table 3**).

### Quality of life of respondents on HAART

This study demonstrated that more than half (53%) of the respondents had good overall QoL.

The highest domain score of QOL was in the level of independence domain with a mean score of 24.91 ± SD 2.99 and lowest in social relations domain with a mean score value of 11.99 ± SD 2.08. In addition to the six domains of QoL, the tool also measures general perceived

**Table 1. Socio-demographic and economic characteristics of quality of life among PLHIV in Jimma zone public hospitals, South-west Ethiopia.**

| Variables | Category | Frequency | percentage |
|---|---|---|---|
| **Residence** | Rural | 95 | 31.7 |
| | Urban | 205 | 68.3 |
| **Sex** | Male | 113 | 37.7 |
| | Female | 187 | 62.3 |
| **Age** | < = 24 | 9 | 3.0 |
| | 25–34 | 95 | 31.7 |
| | 35–44 | 133 | 44.3 |
| | > = 45 | 63 | 21 |
| **Educational status** | < = grade 4 | 107 | 35.7 |
| | Primary(5–8) | 97 | 32.3 |
| | Secondary(8–12) | 53 | 17.7 |
| | Postsecondary | 43 | 14.3 |
| **Employment status** | Government Employed | 50 | 16.7 |
| | Private Business | 117 | 39 |
| | Student/laborer(Unemployed) | 91 | 30.3 |
| | House wife | 42 | 14 |
| **Ethnicity** | Oromo | 165 | 55.0 |
| | Kaffa | 31 | 10.3 |
| | Dawuro, | 27 | 9.0 |
| | Amhara | 57 | 19.0 |
| | Other* | 20 | 6.7 |
| **Marital status** | Married | 27 | 9 |
| | Single | 181 | 60.3 |
| | Separated / Divorced | 49 | 16.3 |
| | Widowed | 43 | 14.3 |
| **Wealth status** | 1$^{st}$ quintile | 100 | 33.3 |
| | 2$^{nd}$ quintile | 104 | 34.7 |
| | 3$^{rd}$ quintile | 96 | 32.0 |
| **Body Mass Index** | Underweight(<18.5 Kg/m$^2$) | 67 | 22.3 |
| | Normal(18.5–24.9 Kg/m$^2$) | 173 | 57.7 |
| | Over weight(25–29.9 Kg/m$^2$) | 46 | 15.3 |
| | Obese(> = 30 Kg/m$^2$) | 14 | 4.6 |
| **Duration on ART** | <36 months | 68 | 22.7 |
| | > = 36 months | 232 | 77.3 |

Other* = Tigre and Gurage

quality of life and health status. About 31.3% of the participants reported to have poor general QoL, whereas 20.7% of respondents reported poor perception of health (**Table 4**).

## Factors associated with poor quality of life among PLWHIV on HAART

In the bivariate logistic regression analysis residence, religion, ethnicity, educational status, age of respondents, employment status, Alcohol frequency, tobacco smoking, khat chewing frequency, wealth status, marital status, Getting support, Getting Emotional, Getting physical care, support from NGO, from government, From Work Place, support from family, Satisfaction of the overall support, and depression were selected candidate variables at P-value < = 0.25.

**Table 2. Behavioral characteristics of respondents with quality of life among PLHIV in Jimma zone public hospitals, South-west Ethiopia.**

| Variable | Category | Frequency | Percentage |
|---|---|---|---|
| Ever use of any substance | Yes | 140 | 46.7 |
| | No | 160 | 53.3 |
| Ever use of alcohol | Yes | 56 | 18.7 |
| | No | 244 | 81.3 |
| Ever use of khat | Yes | 97 | 32.3 |
| | No | 203 | 67.7 |
| Ever use of tobacco | Yes | 60 | 20 |
| | No | 240 | 80 |
| Alcohol frequency | Never | 267 | 89.0 |
| | Once or twice per month | 13 | 4.3 |
| | 4 times per month | 15 | 5.0 |
| | Daily | 5 | 1.7 |
| Chat chewing frequency | Never | 222 | 76.3 |
| | Once or twice per month | 19 | 6.3 |
| | 4 times per month | 32 | 10.7 |
| | Daily | 20 | 6.7 |
| Tobacco smoking frequency | Never | 276 | 92 |
| | Once or twice per month | 13 | 4.3 |
| | 4 times per month | 5 | 1.7 |
| | Daily | 6 | 2 |
| HIV disclosure | Yes | 227 | 76.3 |
| | No | 71 | 23.7 |

In the final model, being government employee, from private business, being in 1st wealth quintile and not obtaining financial support were associated with poor physical health. Being from the 1st wealth quintile, chewing khat 4 times per month, not obtaining support from the government was associated with poor psychological health. Being secondary educational status, 1st wealth quintile, 2nd wealth quintile, and tobacco smoking was associated with poor level of independence. Being separated / divorced, government employee, student, 1st wealth quintile, not obtaining physical care and support, mild depression, moderate depression were associated with social relationship. Being from rural residence, in the 1st wealth quintile, 2nd wealth quintile, not obtaining financial support, was associated with poor environmental domain. Finally not obtaining financial support, and support from family was associated with spiritual domain.

Factors associated with poor physical health includes being government employee AOR 0.33 95%CI (0.15, 0.69), from private business AOR 0.33 95%CI (0.14, 0.79), being 1st quintile AOR 2.44 95%CI (1.16, 5.14), and not obtaining financial support AOR 4.27 95%CI (1.94, 9.42). Being from the 1st quintile AOR 2.06 95%CI (1.13, 3.76), chewing khat 4 times per month AOR 0.32 95%CI (0.15, 0.73), chewing khat daily AOR 0.28 95%CI (0.10, 0.76), not obtaining government support AOR 2.55 95%CI (1.32, 4.92) was associated with poor psychological health. Being secondary educational status AOR 3.56 95%CI (1.13, 11.21), 1st quintile AOR 4.24 95%CI (1.96, 9.21), 2nd quintile AOR 3.79 95%CI (1.83, 7.87), were associated with poor level of independence. Being separated / divorced AOR 0.08 95%CI (0.01, 0.42), government employee AOR 0.17 95% CI (0.05, 0.59), student AOR 0.09 95% CI (0.02, 0.41), 1st quintile AOR 5.31 95%CI (1.69, 16.66), not obtaining physical care and support AOR 3.67 95% CI (1.35, 9.99), mild depression AOR 0.24 95% CI (0.09, 0.67), moderate depression AOR 0.05 95% CI (0.02, 0.16) was associated with social relationship.

**Table 3. Psychosocial characteristics of respondents on quality of life among PLHIV in Jimma zone public hospitals, South-west Ethiopia.**

| Variable | Category | Frequency | Percentage |
|---|---|---|---|
| **Getting support** | Yes | 175 | 58.3 |
| | No | 125 | 41.7 |
| **Getting Emotional support** | Yes | 72 | 24 |
| | No | 228 | 76 |
| **Getting financial support** | Yes | 60 | 20 |
| | No | 240 | 80 |
| **Getting Physical Care and Support** | Yes | 78 | 26 |
| | No | 222 | 74 |
| **Support obtained from friends** | Yes | 19 | 6.3 |
| | No | 281 | 93.7 |
| **Non-Government Organization** | Yes | 14 | 4.7 |
| | No | 286 | 95.3 |
| **Community Based Organization** | Yes | 7 | 2.3 |
| | No | 293 | 97.7 |
| **Religious Based Organization** | Yes | 7 | 2.3 |
| | No | 293 | 97.7 |
| **Government Organization** | Yes | 48 | 16 |
| | No | 252 | 84 |
| **From Work Place** | Yes | 3 | 1 |
| | No | 297 | 99.0 |
| **From Family** | Yes | 115 | 39.3 |
| | No | 185 | 61.7 |
| **Satisfaction of the overall support** | Dissatisfied | 12 | 4.0 |
| | Somehow Dissatisfied | 37 | 12.3 |
| | Very satisfied | 122 | 40.7 |
| | Neither dissatisfied/satisfied | 129 | 43 |
| **Stigma** | Stigmatized | 234 | 78 |
| | Not stigmatized | 66 | 22 |
| **Depression** | Minimal | 113 | 37.7 |
| | Mild | 45 | 15 |
| | Moderate | 63 | 21 |
| | Severe | 79 | 26.3 |

Being from rural residence AOR 0.45 95%CI (0.25, 0.79), in the 1st quintile AOR 5.13 95% CI (2.50, 10.53), 2nd quintile AOR 2.44 95%CI (1.25, 4.76), not obtaining financial support AOR 2.82 95%CI (1.33, 5.98), was associated with Environmental. Not obtaining financial support AOR 2.67 95%CI (1.30, 5.48), not obtaining support from family AOR 0.29 95%CI (0.16, 0.51) was associated with spiritual domain (Table 5).

**Table 4. Mean domain scores of quality of life of respondents on quality of life among PLHIV in Jimma zone public hospitals, South-west Ethiopia.**

| Domains | Cronbach's α | Mean | Standard deviation | Poor Overall qol (%) | Good Overall qol (%) |
|---|---|---|---|---|---|
| Physical health | 0.754 | 13.24 | 1.73 | 154(51.3) | 146(48.7) |
| Psychological health | 0.779 | 14.41 | 1.90 | 145(48.3) | 155(51.7) |
| Social relationship | 0.781 | 11.99 | 2.08 | 116(38.7) | 184(61.3) |
| Environment | 0.763 | 13.75 | 2.45 | 129(43) | 171(57) |
| Level of independence | 0.791 | 24.91 | 2.99 | 124(41.3) | 176(58.7) |
| Spiritual health. | 0.781 | 12.56 | 1.75 | 156(52) | 144(48) |

**Table 5. Independent factors of poor QOL domains among PLHIV on HAART in public hospitals of Jimma zone, Southwest Ethiopia.**

| Variable | Category | PH AOR 95%CI | Psy AOR 95%CI | Ind AOR 95%CI | Soc AOR 95%CI | Envi AOR 95%CI | Spir AOR 95%CI |
|---|---|---|---|---|---|---|---|
| **Place of residence** | Rural | | | | | 0.45(0.25,0.79)* | |
| | Urban | 1.00 | 1.00 | 1.00 | 1.00 | 1.00 | 1.00 |
| **Education status** | < = grade 4 | | | 0.41(0.16,1.05) | | | |
| | Primary(5–8) | | | 0.91(0.34,2.43) | | | |
| | Secondary(8–12) | | | 3.56 (1.13,11.21)* | | | |
| | Post Secondary | 1.00 | 1.00 | 1.00 | 1.00 | 1.00 | 1.00 |
| **Marital status** | Married | 1.00 | 1.00 | 1.00 | 1.00 | 1.00 | 1.00 |
| | Single | | | | 1.02(0.23,4.44) | | |
| | Separated / Divorced | | | | 0.08(0.01,0.42)* | | |
| | Widowed | | | | 0.37(0.07,2.05) | | |
| **Employment** | Government Employed | 0.33(0.15,0.69)* | | | 0.17(0.05,0.59)* | | |
| | Private Business | 0.33(0.14,0.79)* | | | 0.46(0.11,1.87) | | |
| | Student | 0.59(0.23,1.55) | | | 0.09(0.02,0.41)* | | |
| | House wife | 1.00 | 1.00 | 1.00 | 1.00 | 1.00 | 1.00 |
| **Wealth** | 1st quintile | 2.44(1.16,5.14)* | 2.06(1.13,3.76)* | 4.24(1.96,9.21)** | 5.31 (1.69,16.66)* | 5.13 (2.50,10.53)** | |
| | 2nd quintile | 1.94(0.99,3.82) | 1.23(0.63,1.99) | 3.79(1.83,7.87)** | 2.57(0.93,7.11) | 2.44(1.25,4.76)* | |
| | 3rd quintile | 1.00 | 1.00 | 1.00 | 1.00 | 1.00 | 1.00 |
| **Kat frequency** | Never | 1.00 | 1.00 | 1.00 | 1.00 | 1.00 | 1.00 |
| | Once or twice per month | | 1.34(0.49,3.63) | | | | |
| | 4 times per month | | 0.32(0.15,0.73)* | | | | |
| | Daily | | 0.28(0.10,0.76)* | | | | |
| **Financial Support** | Yes | 1.00 | 1.00 | 1.00 | 1.00 | 1.00 | 1.00 |
| | No | 4.27 (1.94,9.42)** | | | | 2.82(1.33,5.98)* | 2.67(1.30,5.48)* |
| **Physical support** | Yes | 1.00 | 1.00 | 1.00 | 1.00 | 1.00 | 1.00 |
| | No | | | | 3.67(1.35,9.99)* | | |
| **Support from Government** | Yes | 1.00 | 1.00 | 1.00 | 1.00 | 1.00 | 1.00 |
| | No | | 2.55(1.32,4.92)* | | | | |
| **Support from family** | Yes | | | | | | 0.29(0.16,0.51)** |
| | No | | | | | | |
| **Depression** | Minimal | 1.00 | 1.00 | 1.00 | 1.00 | 1.00 | 1.00 |
| | Mild | | | | 0.24(0.09,0.67)* | | |
| | Moderate | | | | 0.05 (0.02,0.16)** | | |
| | Severe | | | | 0.78(0.23,2.62) | | |

AOR = adjusted odds ratio, CI = confidence interval

*$P < 0.05$

**$P < 0.001$, 1 = reference category, PH = physical health, Psy = psychological health, Soc = Social relationship, Env = Environment, Ind = level of independence, Spir = Spiritual health. The Hosmer-Lemeshow goodness-of-fit test statistic is greater than 0.05 for all the models.1.00 = reference

## Discussion

Measuring and evaluating the of QoL of PLWHIV on HAART becomes an increasingly growing concern as it aims to improve the degree of excellence in a person's life and contributes to satisfaction and happiness of the person and benefits society [14, 15]. In this study, more than

half of the respondents had good overall QoL. This shows consistent finding with a study conducted in Southern Ethiopia with the overall QOL (52.9%) [39], and lower than a study in Georgia (63.7%) [40]. However, our study showed a higher overall QOL than a study conducted in Bahir-dar [21], in Health Facilities of Jimma Town [41], and in China [25].

Our study demonstrated that the level of independence domain was highest with a mean score of 24.91 ± SD 2.99 which showed higher domain score as compared to studies conducted elsewhere [25, 40, 41]. However, the lowest domain scores in social relations domain with a mean score value of 11.99± SD 2.08 shows consistent finding with studies conducted in different settings of Ethiopia [25, 39, 41], and in Georgia [40]. The differences in the domain and overall mean scores of QOL could be attributed to the socio demographic, economic and service quality differences across the study participants and settings.

In this study several factors have contributed to the different domains of poor QOL. The lower wealth index has been associated with almost all domain scores of poor QOL except the spiritual domain. Similarly, employment status and lack of obtaining psychosocial support have been associated with the outcome variable.

The factors found to have significant influence on physical health of quality of life domain in multiple logistic regressions were being government employee, from private business, being in 1st wealth quintile and not obtaining financial support. This is supported by a study conducted in Mekelle Town, Northern Ethiopia [23], in Jimma town health facilities [41]. A study from Ghana reported that informal occupational status was associated with poor QOL [42]. A study conducted in Central zone of Tigray, Northern Ethiopia revealed that being from the second quintile was associated with malnutrition which on the other hand verified that malnourished individuals are more likely to have poor QOL [43].

Factors associated with poor psychological health includes being in the 1st wealth quintile, chewing khat at weekly intervals and not obtaining support from the government. Consistent with similar studies [23, 41], a study from India observed that participants with higher SES showed better psychosocial QOL domain scores [44]. A study from Northern Ethiopia identified that substance use was associated with ART Non adherence [28]. Although this study reported that chewing chat at weekly intervals had protective effect, a study conducted in a similar setting verified that Khat use was associated with mental distress and missing of at least on ART drug which could lead to poor psychological health of PLWHIV [45].

Educational status and lower wealth quintiles was associated with poor level of independence. Unlike to the present study, a person with lower education status was more likely to have poor general QoL [39, 40, 46]. On the other hand being from the lower wealth quintiles was associated with poor level of independence which shows consistent finding with a study conducted in Coastal South India [44].

Being separated / divorced, government employee, student, 1st wealth quintile, not obtaining physical care and support, depression, were associated with poor social relationship QOL. Our study is comparable with studies conducted elsewhere that the likelihood of poor QOL is much higher among individuals with less established economic status and employment level [19, 21, 22]. Similarly, an exploratory study conducted in Jordan showed that unemployment low income, and single statuses (separated, divorced or widowed) were connected with poor HRQoL [47]. Depression was associated with poor quality of life which is consistent with studies done in a similar setting of Addis Ababa, South India, by Pederson KK, and China, followed by high levels of HIV stigma [18, 30, 46–48]. Studies supported that obtaining social support was associated with better QOL [18, 22, 25, 30, 41, 49]. A study from Beijing also identified that family support along with no or minimal discrimination was found to contribute to QOL among people infected with HIV [50].

Being from rural residence, lower wealth quintile, not obtaining financial support, was associated with poor environmental domain. Unlike this study, it was shown that rural residence was associated with poor environmental domain [22]. However, lower wealth quintile and not obtaining financial support was associated with poor QoL [21, 22, 44, 50, 51].

Not obtaining financial and family was associated with spiritual domain. This is in line with studies conducted across the world [17, 18, 22, 25, 30, 40]. However, our study contradicts with a study China that patients' overall QOL scores were positively associated with having received family support [48]. This is mainly because social support is crucial for adherence to ART drugs which in turn improves the patients' quality of lives [10, 25, 45, 50].

Finally, our study found nearly 80% and 26.3% of the study participants were stigmatized and severely depressed. Although stigma did not show significant association of with the outcome variables, studies demonstrated that a high perceived stigma, severe self-stigma, and those with severe disclosure concerns were strongly associated with poor quality of life [18, 21, 22, 39, 48, 51].

## Limitations of the study

Being a cross-sectional study, causal inference cannot be made between QoL andindependent variables especially the relationship between QOL in our participants was not compared to similar analysis of the same variables. Since the QOL questionnaire has some sensitive issues like behavioral variables, social desirability bias might be induced. The measurement of QOL was not consistent across different literatures, therefore this study used researches conducted to measure health related QOL and QOL interchangeably for comparison. Despite these limitations, the study has strengths such as training of data collectors helped to reduce social desirability bias; identification of eligible and non-eligible participants was also performed by controlling possible confounders.

## Conclusion

More than half of the study participants had good overall QOL with the highest domain score in level of independence and lowest in social relations. Lower wealth index has been associated with almost all poor domain scores of QOL except spiritual domain. Being government employee and private business, being in 1st wealth quintile and not obtaining financial support were associated with poor physical health. Being from the 1st wealth quintile, chewing khat 4 times per month, not obtaining support from the government was associated with poor psychological health. Being secondary educational status and 1st and 2nd wealth quintiles were associated with poor level of independence. Being separated / divorced, government employee, student, 1st wealth quintile, not obtaining physical care and support, mild depression moderate depression were associated with poor social relationship. Being from rural residence, 1st and 2nd wealth quintiles and not obtaining financial support were associated with poor environmental domain. Finally not obtaining financial and family support was associated with poor spiritual domain.

Therefore, to curb the consequences of poor QOL, it will become all the most important to develop effective strategies, policies and programs targeting people living with HIV. Emphasis should be given to the socio-economic factors that affect their QOL on HAART. Professional counseling and guidance with life skill packages should be strengthened to cope up with adverse behavioral factors. Finally, psychosocial support should be provided from all responsible bodies.

## Supporting information

**S1 File. Operational definitions and definition of terms.**
(DOCX)

**S1 Questionnaire. English version questionnaire.**
(DOCX)

**S2 Questionnaire. Afan Oromo version questionnaire.**
(DOCX)

**S3 Questionnaire. Amharic version questionnaire.**
(DOCX)

**S1 Data. Data set on behavioral and psychosocial factors.**
(XLSX)

## Acknowledgments

Study participants are greatly acknowledged for the information they provided. Finally, we would like to share our acknowledgment to the research square.

## Author Contributions

**Conceptualization:** Abreha Addis Gesese, Yitages Getachew Desta, Endale Zenebe Behire.

**Data curation:** Abreha Addis Gesese, Yitages Getachew Desta, Endale Zenebe Behire.

**Formal analysis:** Abreha Addis Gesese, Yitages Getachew Desta, Endale Zenebe Behire.

**Investigation:** Abreha Addis Gesese.

**Methodology:** Abreha Addis Gesese.

**Validation:** Abreha Addis Gesese.

**Writing – original draft:** Abreha Addis Gesese.

**Writing – review & editing:** Abreha Addis Gesese, Yitages Getachew Desta, Endale Zenebe Behire.

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
