## [Decision Letter · Decision Letter 0]

18 Oct 2021

PGPH-D-21-00472

Behavioral and psychosocial factors of Quality of life among Adult people living with HIV on Highly Active Antiretroviral Therapy, in public hospitals of South West Ethiopia, 2018: A Case control study

Dear Dr. Gesese,

Thank you for submitting your manuscript to PLOS Global Public Health. After careful consideration, we feel that it has merit but does not fully meet PLOS Global Public Health’s publication criteria as it currently stands. Therefore, we invite you to submit a revised version of the manuscript that addresses the points raised during the review process.

We look forward to receiving your revised manuscript.

Kind regards,

Olatunji O Adetokunboh, MD, PhD

Academic Editor

Journal Requirements:

1. Please include additional information regarding the survey or questionnaire used in the study and ensure that you have provided sufficient details that others could replicate the analyses. For instance, if you developed a questionnaire as part of this study and it is not under a copyright more restrictive than CC-BY, please include a copy, in both the original language and English, as Supporting Information.

2. In the online submission form, you indicated that "The data sets used and/or analyzed during the current study are available from the corresponding author of the research on reasonable request." All PLOS journals now require all data underlying the findings described in their manuscript to be freely available to other researchers, either 1. In a public repository, 2. Within the manuscript itself, or 3. Uploaded as supplementary information.

3. State what role the funders took in the study. If the funders had no role in your study, please state: “The funders had no role in study design, data collection and analysis, decision to publish, or preparation of the manuscript.”

Additional Editor Comments (if provided):

Reviewers' comments:

Reviewer's Responses to Questions

**Comments to the Author**

1. Does this manuscript meet PLOS Global Public Health’s publication criteria? Is the manuscript technically sound, and do the data support the conclusions? The manuscript must describe methodologically and ethically rigorous research with conclusions that are appropriately drawn based on the data presented.

Reviewer #1: No

Reviewer #2: Partly

Reviewer #3: Partly

2. Has the statistical analysis been performed appropriately and rigorously?

Reviewer #1: No

Reviewer #2: No

Reviewer #3: Yes

3. Have the authors made all data underlying the findings in their manuscript fully available (please refer to the Data Availability Statement at the start of the manuscript PDF file)?

Reviewer #1: No

Reviewer #2: No

Reviewer #3: Yes

4. Is the manuscript presented in an intelligible fashion and written in standard English?

Reviewer #1: No

Reviewer #2: No

Reviewer #3: No

5. Review Comments to the Author

Reviewer #1: The authors submitted a manuscript reporting a study to determine the association of behavioral and psychosocial factors with the Quality of life among adult people living with HIV on Highly Active Antiretroviral Therapy (HAART) in selected public hospitals of South West Ethiopia. Whilst this is an important topic within the scope of this journal, the authors did not provide adequate scientific justifications for the study design, sample size calculation and stastical analysis. Also, the conclusions drawn from the study were not reflective of the study findings. More importantly, the manuscript was not well written in an acceptable standard English as it contains a lot of syntax errors and incomplete statements that made it difficult to understand or follow the lines of thoughts of the authors. I have highlighlighted some major concerns, while the minor comments were indicated within within the body of the manuscript:

1. Abstract: Given the title of this manuscript, it would be extremely important for the authors to include the name of the tool they used to assess the Quality of Life of the study participants.

How many of the participants were severely depressed among the cases and the controls? What tool was used to assess the study participants to arrive at this classification?

2. Introduction: HIV/AIDS is no more a fatal illness as it has become a chronic manageable communicable disease, with advent of HAART, even in low-income countries Authors should check the follow recent publications: https://innovation.org/diseases/infectious/hiv-aids/hiv-aids-acute-fatal-disease-chronic-manageable-condition; 3.https://www.ncbi.nlm.nih.gov/pmc/articles/PMC3122586/

3.Authors should consider using a more recent data on national seroprevalence of HIV in Ethiopia because the prevalence of 1.1% does not support the claim of Ethiopia being a country with high burden of HIV. Also, the national prevalence cited here was that of 2017. Data presenting the trends of the national prevalence of HIV in Ethiopia may be useful for this context: https://knoema.com/atlas/Ethiopia/HIV-prevalence

4. Authors should indicate whether this tool has been previously used in this setting or elsewhere in Ethiopia. Authors should also include the reliability and validity of the tool. Authors should indicate whether the English version of the

instrument was used in the study, or a translated versions in local languages of the study participants were used. If yes, what were the reliability and validity indices of the translated versions of the instrument. Authors should check this link: https://ijmhs.biomedcentral.com/articles/10.1186/s13033-016-0062-x

5.his is a cumbersome definition of case and control definition for this study. WHOQoL-HIV BREF. What were the mean scores of WHOQoL-HIV BREF? How long were the PLWHIV enrolled in the HAART? Did the authors assess the quality of life of the PLWHIV before assigning recruiting them into either as a case or control in this study? These are difficult to understand from the authors' definition. Authors clarify these.

6. Authors should support this sentence with appropriate references because ref 31 cited was not a case-control study. It was a cross-sectional hospital based study in Nigeria which has one of the highest burden of HIV in the world, after India and South-Africa. Using the prevalence obtained from a hospital-base cross-sectional study to determine the sample size of case-control study in a setting that is not similar to Nigeria, is not only inappropriate but also illogical.

7. Was the questionnaire administered in English or in the local languages? What was the literacy level in the study population? If administered in local languages, was the questionnaire translated to the local languages through forward and backward translation? If not translated, how did the interviewers ensure consistency in administering the questionnaire to the study participants.

8.Authors should provide the justifications of p<0.25 for this study.

9. The references were not consistently cited. Authors should ensure the referencing styling was consistent, they should also provide missing accessed dated and correct URLs.

Reviewer #2: PGPH-D-21-00472

Thank you for the opportunity to review this manuscript which describes the behavioural and psychological factors affecting the quality of life of people living with HIV in Ethiopia. Despite the successful implementation of anti-retroviral therapy in many countries, PLHIV do face challenges that affect their quality of life, and it is important that the body of science in this area be expanded. Here are my comments to the authors.

In brief, the authors need to justify the need for this study clearly in the introduction section, the design used does not appear appropriate but if the authors want to keep it as a case-control design, then they must justify it based on principle and their study procedures. The results section requires significant improvement, this section must start with a table showing the QoL assessments, so readers are able to see how the cases and controls scored on the assessments. Tables need to be numbered sequentially and the multivariable assessment need to be redone for BMI

Abstract

1. Please define your cases and controls clearly.

2. Multivariate regression is different from multivariable regression. Did the authors assess risk factors for multiple outcomes and risk factors in the same model – that is what multivariate regression is

3. Please give a brief description of the characteristics of the participants

Introduction

1. Please copy edit and perhaps get an English editor the whole manuscript. There are many instances where an article “A” or “The” is missing at the start of sentences, where punctuations are missing, and where grammar needs to be revised.

2. Please use scientific terms. The first sentence on page 4 needs to be specific about what the authors mean by “suffer”

“Despite different strategies provoked and tremendous efforts done to halt the spread and

consequences of HIV/AIDS both at the global as well as national level, PLWHIV continued to

suffer.”

3. The introduction introduces the concept of QL and HIV but does not show clearly where the gaps in current literature are and what gaps the current study intends to fulfil

Methods

1. There is no clear rationale why a case-control designed was used. Is poor QL a scarce outcome? It appears the authors samples by exposure and then assessed the outcome (using WHOQOL-HIV BREF).Or both at the same time – a classic cross sectional study, especially in view that there was no matching done

2. Please do not use the personal pronoun “I” as below

“For the purpose of this study, i defined quality of life as personal evaluation of how things have

been going for one self, and as how the individual‘s wellbeing may be impacted over time by a

disease, a disability, or a disorder.”

3. Please provide references for the definitions on page 7

Results

1. By a response rate of 100%, do the authors mean that all the participants they approached agreed to take part? How many eligible individuals were there? A flow chart would be highly helpful

2. Please present the results of the WHOQOL-HIV BREF assessments

3. Please be consistent on decimal places. If you choose to use 1dp, then do so throughout, except for counts

4. Tables should be numbered sequentially. Is there a reason to start with Table 2?

5. In Table 2, what is the importance of distinguishing between <grade 4="" 5-8.="" and="" are="" different="" education="" ethiopian="" grades="" in="" key="" stages="" system="" the="" these="">6. Please put p-values in the Tables if you wish to then use those p-values as criteria for selection into multivariable models

7. Table 5 – this is multivariable and not multivariate analysis

8. Why is “obese” used as the base comparison for BMI categories – pleased use “normal”

9. Please do not use bold for “significant results”

10. The model needs to be redone. Also, would be useful for the authors to submit the data underlying the study so reviewers (and readers) can run the models themselves. We are in a situation where Open Science has become more important than ever, and “data available from authors upon reasonable request” is no longer sufficient. However, this is my personal opinion and the authors should be guided by the journal policy</grade>

Reviewer #3: The authors conducted a case-control study to identify factors (data collected via a survey) associated with low perceived quality of life among people living with HIV in Ethiopia.

The manuscript requires revision to ensure correct grammar and syntax in English, as well as formatting (particularly in the tables).

Participant recruitment is not clearly explained. A survey was conducted and survey data were used to define participants as cases and controls. But the final analysis data set matches exactly the sample size calculation, which does not seem to be representative of the overall distribution of QoL scores (1 case <mean 3="" :="" controls="">mean). How many total people were surveyed? How many survey participants who met the criteria for a case or control were then not included in the analysis? How was consecutive recruitment maintained in a 1:3 case-control ratio, when the definition of case and control is a mean survey score? A flow diagram detailed participant recruitment in the survey and inclusion in the analysis data set would be helpful.

Where did the estimated exposure to psychosocial factors in the sample size calculation come from?

Definitions could be included as supplementary material, rather than in the methods section.

Most of the findings from the multi-variate analysis (e.g. people experiencing food insecurity, disease stigma and depression have a low perceived quality of life) do not add anything new to the literature. The discussion confirms that these findings are not novel.</mean>

6. PLOS authors have the option to publish the peer review history of their article (what does this mean?). If published, this will include your full peer review and any attached files.

**Do you want your identity to be public for this peer review?** For information about this choice, including consent withdrawal, please see our Privacy Policy.

Reviewer #1: **Yes: **Dr Muhammed Afolabi, London School of Hygiene & Tropical Medicine, UK

Reviewer #2: No

Reviewer #3: No

---

## [Decision Letter · Decision Letter 1]

9 Feb 2022

PGPH-D-21-00472R1

Behavioral and psychosocial factors of Quality of life among Adult people living with HIV on Highly Active Antiretroviral Therapy, in public hospitals of South West Ethiopia, 2018: A Case-control study

Dear Dr. Gesese,

Thank you for submitting your manuscript to PLOS Global Public Health. After careful consideration, we feel that it has merit but does not fully meet PLOS Global Public Health’s publication criteria as it currently stands. Therefore, we invite you to submit a revised version of the manuscript that addresses the points raised during the review process.

We look forward to receiving your revised manuscript.

Kind regards,

Olatunji O Adetokunboh, MD, PhD

Academic Editor

Additional Editor Comments (if provided):

Reviewers' comments:

Reviewer's Responses to Questions

**Comments to the Author**

1. If the authors have adequately addressed your comments raised in a previous round of review and you feel that this manuscript is now acceptable for publication, you may indicate that here to bypass the “Comments to the Author” section, enter your conflict of interest statement in the “Confidential to Editor” section, and submit your "Accept" recommendation.

Reviewer #2: All comments have been addressed

Reviewer #4: All comments have been addressed

Reviewer #5: (No Response)

2. Does this manuscript meet PLOS Global Public Health’s publication criteria? Is the manuscript technically sound, and do the data support the conclusions? The manuscript must describe methodologically and ethically rigorous research with conclusions that are appropriately drawn based on the data presented.

Reviewer #2: No

Reviewer #4: Yes

Reviewer #5: Partly

3. Has the statistical analysis been performed appropriately and rigorously?

Reviewer #2: No

Reviewer #4: Yes

Reviewer #5: Yes

4. Have the authors made all data underlying the findings in their manuscript fully available (please refer to the Data Availability Statement at the start of the manuscript PDF file)?

Reviewer #2: Yes

Reviewer #4: Yes

Reviewer #5: Yes

5. Is the manuscript presented in an intelligible fashion and written in standard English?

Reviewer #2: Yes

Reviewer #4: Yes

Reviewer #5: No

6. Review Comments to the Author

Reviewer #2: Review - PGPH-D-21-00472

Thank you to the authors for the revised manuscript. Although the authors have addressed some of the concerns I raised earlier, this paper is still not ready for publication. I have detailed some concerns below, in the attached document

Reviewer #4: Thank you addressing most of the comments

Reviewer #5: (No Response)

7. PLOS authors have the option to publish the peer review history of their article (what does this mean?). If published, this will include your full peer review and any attached files.

**Do you want your identity to be public for this peer review?** For information about this choice, including consent withdrawal, please see our Privacy Policy.

Reviewer #2: **Yes: **Tawanda Chivese

Reviewer #4: No

Reviewer #5: No

---

## [Decision Letter · Decision Letter 2]

19 May 2022

PGPH-D-21-00472R2

Behavioral and psychosocial factors of Quality of life among Adult people living with HIV on Highly Active Antiretroviral Therapy, in public hospitals of South West Ethiopia, 2018: A Case-control study

Dear Dr. Gesese,

Thank you for submitting your manuscript to PLOS Global Public Health. After careful consideration, we feel that it has merit but does not fully meet PLOS Global Public Health’s publication criteria as it currently stands. Therefore, we invite you to submit a revised version of the manuscript that addresses the points raised during the review process.

We look forward to receiving your revised manuscript.

Kind regards,

Olatunji O Adetokunboh, MD, PhD

Academic Editor

Journal Requirements:

Additional Editor Comments (if provided):

Reviewers' comments:

Reviewer's Responses to Questions

**Comments to the Author**

1. If the authors have adequately addressed your comments raised in a previous round of review and you feel that this manuscript is now acceptable for publication, you may indicate that here to bypass the “Comments to the Author” section, enter your conflict of interest statement in the “Confidential to Editor” section, and submit your "Accept" recommendation.

Reviewer #2: All comments have been addressed

Reviewer #4: All comments have been addressed

2. Does this manuscript meet PLOS Global Public Health’s publication criteria? Is the manuscript technically sound, and do the data support the conclusions? The manuscript must describe methodologically and ethically rigorous research with conclusions that are appropriately drawn based on the data presented.

Reviewer #2: No

Reviewer #4: Yes

3. Has the statistical analysis been performed appropriately and rigorously?

Reviewer #2: No

Reviewer #4: Yes

4. Have the authors made all data underlying the findings in their manuscript fully available (please refer to the Data Availability Statement at the start of the manuscript PDF file)?

Reviewer #2: No

Reviewer #4: Yes

5. Is the manuscript presented in an intelligible fashion and written in standard English?

Reviewer #2: No

Reviewer #4: No

6. Review Comments to the Author

Reviewer #2: Dear Editor, the authors have responded well to some of the issues I raised earlier. However, the design and analysis of this paper is still not adequate, and the paper is not suitable for publication.

Minor issues

This paper requires proof reading by an English expert as there are many grammatical and language errors.

Major issues

Study design

The authors response is noted. I am still not convinced that the case control design is the correct design for this study. This is worsened by an inadequate case definition using a cut-off based on the mean in their sample. This is not a correct definition of “poor” quality of life.

1. Can the authors please confirm that they coded the WHOQOL-HIV BRIEF correctly as in the WHO manual https://apps.who.int/iris/rest/bitstreams/109786/retrieve

2. Where are the data for the WHOQOL-HIV BRIEF and the other data in the study?

3. The WHOQOL-HIV BRIEF does not provide cut-offs, which could have been useful for the authors to use to define their cases and controls. In the absence of validated cut-offs, the authors are advised to analyse the QoL as a continuous outcome, in a cross-sectional design rather than a case control. An example of this approach is here https://www.tandfonline.com/doi/full/10.1080/09540121.2016.1146209

If the authors choose to continue with a cross sectional design, the best way to do so is to define their cases using the extremes (25th percentile or lower) as these are individuals who most likely have significant lower QoL than the others in the sample.

Reviewer #4: Thank you for addressing all necessary comments.

7. PLOS authors have the option to publish the peer review history of their article (what does this mean?). If published, this will include your full peer review and any attached files.

**Do you want your identity to be public for this peer review?** For information about this choice, including consent withdrawal, please see our Privacy Policy.

Reviewer #2: **Yes: **Tawanda Chivese

Reviewer #4: No

---

## [Decision Letter · Decision Letter 3]

8 Jul 2022

Behavioral and psychosocial factors of Quality of life among Adult people living with HIV on Highly Active Antiretroviral Therapy, in public hospitals of Southwest Ethiopia

PGPH-D-21-00472R3

Dear Mr Gesese,

We are pleased to inform you that your manuscript 'Behavioral and psychosocial factors of Quality of life among Adult people living with HIV on Highly Active Antiretroviral Therapy, in public hospitals of Southwest Ethiopia' has been provisionally accepted for publication in PLOS Global Public Health.

Best regards,

Leonardo Martinez

Academic Editor

Reviewer Comments (if any, and for reference):

Reviewer's Responses to Questions

**Comments to the Author**

1. If the authors have adequately addressed your comments raised in a previous round of review and you feel that this manuscript is now acceptable for publication, you may indicate that here to bypass the “Comments to the Author” section, enter your conflict of interest statement in the “Confidential to Editor” section, and submit your "Accept" recommendation.

Reviewer #2: All comments have been addressed

2. Does this manuscript meet PLOS Global Public Health’s publication criteria? Is the manuscript technically sound, and do the data support the conclusions? The manuscript must describe methodologically and ethically rigorous research with conclusions that are appropriately drawn based on the data presented.

Reviewer #2: Yes

3. Has the statistical analysis been performed appropriately and rigorously?

Reviewer #2: Yes

4. Have the authors made all data underlying the findings in their manuscript fully available (please refer to the Data Availability Statement at the start of the manuscript PDF file)?

Reviewer #2: No

5. Is the manuscript presented in an intelligible fashion and written in standard English?

Reviewer #2: Yes

6. Review Comments to the Author

Reviewer #2: the authors have addressed my concerns

7. PLOS authors have the option to publish the peer review history of their article (what does this mean?). If published, this will include your full peer review and any attached files.

**Do you want your identity to be public for this peer review?** For information about this choice, including consent withdrawal, please see our Privacy Policy.

Reviewer #2: **Yes: **tawanda chivese
